# COLLABORATE TO DEFEND AGAINST ADVERSARIAL ATTACKS

## ABSTRACT

Adversarially robust learning methods require invariant predictions to a small neighborhood of its natural inputs, thus often encountering insufficient model capacity. Learning multiple sub-models in an *ensemble* can mitigate this insufficiency, further improving both generalization and robustness. However, an *ensemble* still wastes the limited capacity of multiple models. To optimally utilize the limited capacity, this paper proposes to learn a *collaboration* among multiple sub-models. Compared with the *ensemble*, the *collaboration* enables the possibility of correct predictions even if there exists a single correct sub-model. Besides, learning a collaboration could enable every sub-model to fit its vulnerability area and reserve the rest of the sub-models to fit other vulnerability areas. To implement the idea, we propose a collaboration framework—$CDA^2$ the abbreviation for Collaborate to Defend against Adversarial Attacks. $CDA^2$ could effectively minimize the vulnerability overlap of all sub-models and then choose a representative sub-model to make correct predictions. Empirical experiments verify that $CDA^2$ outperforms various ensemble methods against black-box and white-box adversarial attacks.

## 1 INTRODUCTION

Safety-critical applications (such as in medicine and finance) require the *adversarial robustness* of deep models (Goodfellow et al., 2015; Szegedy et al., 2014). An adversarially robust learning method requires invariant predictions to a small neighborhood of its natural inputs, thus often encountering insufficient model capacity (Zhang et al., 2021; Yu et al., 2021a). This limits the further improvement of robustness and has the undesirable degradation of generalization (Madry et al., 2018).

Learning multiple sub-models in an ensemble (Breiman, 1996; Freund et al., 1996) can mitigate this insufficiency (Pang et al., 2019; Kariyappa & Qureshi, 2019; Yang et al., 2020a). Remarkably, Pang et al. (2019), Kariyappa & Qureshi (2019) and Yang et al. (2020a) minimized the vulnerability overlaps between each pair of sub-models and improved both robustness and generalization over a single model.

However, an ensemble wastes the limited capacity of multiple models. In the example of three sub-models (see Figure 1(b)), the adversarial input that lies in the black areas can fool the ensemble successfully, i.e., more than half of sub-models must correctly classify the adversarial input. Therefore, the ensemble's voting-based strategy excludes the possibility that *true predictions remain with the minority*. Besides, learning an ensemble requires more than half of the sub-models to fit the same vulnerability areas, which leaves the following question unanswered whether we could only leverage a single sub-model to fit a vulnerability area and reserve the rest of the sub-models to fit other vulnerability areas.

To optimally utilize the limited capacity, this paper proposes to *learn a collaboration* among multiple sub-models. As shown in Figure 1(c), the adversarial input that lies in the vulnerability overlaps of all sub-models can undoubtedly fool the collaboration. Compared with the ensemble in Figure 1(b)), collaboration enables the possibility of correct predictions even if there exists a single correct sub-model merely. Besides, learning a collaboration could enable every sub-model to fit its vulnerability area, which could collectively fix broader vulnerability areas than the ensemble does. Then, sub-models could collaboratively choose trustworthy ones to make the final predictions.

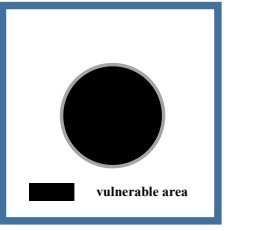 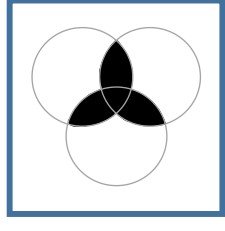 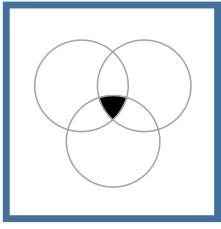

(a) Single model      (b) Ensemble      (c) Collaboration

Figure 1: Illustrations of the vulnerability area of (a) Single model (b) Ensemble, and (c) Collaboration. The black area represents the vulnerability area in which the model is undoubtedly fooled.

To realize the idea, we propose a collaboration framework—Collaborate to Defend against Adversarial Attacks (CDA$^2$) (Algorithms 1 and 2). In CDA$^2$, each sub-model has dual heads: one outputs a vector of predicted probability $f_\theta(\cdot)$; another outputs a scalar that measures posterior probability density (PPD) of the prediction. In the training phase, given a natural or adversarial input $x$, each sub-model chooses an easy one(s) to feed itself. The PPD head is meanwhile updated by comparing the predicted probability on the true label—$f_\theta^y(\cdot)$ (a scalar). In the inference phase, given an input, CDA$^2$ chooses a sub-model with the largest PPD value as the representative to output the prediction.

We highlight our key contributions as follows.

- We provide a new perspective on learning multiple sub-models for defending against adversarial attacks. We theoretically show the collaboration makes better decisions than the ensemble, which implies collaboration may fix broader vulnerability areas.

- We propose a novel collaboration framework—CDA$^2$ (see Section 3.2). In the training phase, CDA$^2$ could effectively minimize the vulnerability overlap of all sub-models; In the inference phase, CDA$^2$ could effectively choose a representative sub-model to make correct predictions. We also provide a comprehensive analysis illustrating the rationale of CDA$^2$.

- Empirical experiments verify that CDA$^2$ outperforms various ensemble methods against black-box and white-box adversarial attacks.

## 2 RELATED WORKS

**Adversarial attack** Adversarial attacks aim to craft the human-imperceptible adversarial input to fool the deep models. Adversarial attacks could be roughly divided into white-box attacks in which the adversary is fully aware of the model's structures (Goodfellow et al., 2015; Moosavi-Dezfooli et al., 2016; Carlini & Wagner, 2017b; Chen et al., 2018; Athalye et al., 2018; Xiao et al., 2018; Zheng et al., 2019; Wong et al., 2019; Mopuri et al., 2019; Alaifari et al., 2019; Sriramanan et al., 2020; Wu et al., 2020b; Croce & Hein, 2020; Yu et al., 2021b) and black-box attacks in which the deep models are treated as black boxes to the adversary (Cheng et al., 2019; 2020; Wu et al., 2020a; Chen et al., 2020a; Li et al., 2020a; Rahmati et al., 2020; Yan et al., 2021b; Hendrycks et al., 2021; Dong et al., 2018; Xie et al., 2019). This paper focuses on building effective defense and select both white-box and black-box attack methods as our robustness evaluation metrics.

**Adversarial defense** Defending adversarial attacks is a challenging task and researchers have proposed various solutions. *Certified defense* tries to learn provably robust deep models against norm-bounded (e.g., $\ell_2$ and $\ell_\infty$) perturbations (Wong & Kolter, 2018; Tsuzuku et al., 2018; Weng et al., 2018; Mirman et al., 2018; Hein & Andriushchenko, 2017; Lécuyer et al., 2019; Xiao et al., 2019; Cohen et al., 2019; Balunovic & Vechev, 2020a; Zhang et al., 2020a; Singla & Feizi, 2020; Balunovic & Vechev, 2020b; Zou et al., 2021). *Empirical defense* leverages adversarial data to build effective defense such as *adversary detection* (Metzen et al., 2017; Li & Li, 2017; Carlini & Wagner, 2017a; Tian et al., 2018; Ma et al., 2018b; Lee et al., 2018; Pang et al., 2018; Smith & Gal, 2018; Roth et al., 2019; Liu et al., 2019; Yin & Rohde, 2020; Sperl et al., 2020; Cohen et al., 2020; Sheikholeslami et al., 2021; Chen et al., 2021a; Yang et al., 2020b; Qin et al., 2020; Tian et al., 2021; Wu et al., 2021) and *adversarial training* (AT), in which AT stands out as the most effective defense. Researchers have investigated various aspects of AT, such as improving AT's robustness or generalization (Madry et al., 2018; Yan et al., 2018; Wu et al., 2018; Cai et al., 2018; Najafi et al.,

2019; Alayrac et al., 2019; Carmon et al., 2019; Farnia et al., 2019; Song et al., 2019; Zhang et al., 2019b; Wang et al., 2019; Tramèr & Boneh, 2019; Zhang & Wang, 2019; Stutz et al., 2020; Pang et al., 2020; Gan et al., 2020; Dong et al., 2020; Zhang et al., 2020b; Chen et al., 2020b; Song et al., 2020; Ding et al., 2020; Wang et al., 2020b; Zhang et al., 2021), fixing AT's undesirable robust overfitting (Rice et al., 2020; Chen et al., 2021b), improving AT's training efficiency (Zhang et al., 2019a; Shafahi et al., 2019; Zheng et al., 2020; B.S. & Babu, 2020; Andriushchenko & Flammarion, 2020; Wong et al., 2020), understanding/interpreting AT's unique traits (Nakkiran, 2019; Yin et al., 2019; Gao et al., 2019; Cranko et al., 2019; Zhang et al., 2019c; Liu et al., 2020; Roth et al., 2020; Wang et al., 2020a; Zhang et al., 2020c; Li et al., 2020b; Zou et al., 2021; Mehrabi et al., 2021; Xu et al., 2021), etc. Besides, researchers have alao actively investigated robust-structured models (Cisse et al., 2017; Xie et al., 2020; Moosavi-Dezfooli et al., 2019; Xie & Yuille, 2020; Yan et al., 2021a; Du et al., 2021; Pang et al., 2021). Nevertheless, the above research thoroughly investigated a single model; this paper focuses on the collaboration among multiple models for adversarial defense.

**Ensemble methods for adversarial robustness** The most relevant studies are the ensemble methods. Ensemble methods such as bagging (Breiman, 1996) and boosting (Freund et al., 1996) have been investigated for significantly improving the model's generalization. Motivated by the benefits of ensemble methods in improving generalization, researchers introduced an ensemble to improve the model robustness (Yang et al., 2020a; Kariyappa & Qureshi, 2019; Pang et al., 2019; Tramèr et al., 2018). Tramèr et al. (2018) proposed to reduce the adversarial transferability by training a single model with adversarial examples from multiple pretrained sub-models. Pang et al. (2019) introduce a regularization method—ADP—to encourage high diversity in the non-maximal predictions of sub-models. Kariyappa & Qureshi (2019) improved the ensemble diversity by maximizing the introduced cosine distance between the gradients of sub-models with respect to the input. Yang et al. (2020a) proposed to distill non-robust features in the input and diversify the adversarial vulnerability. These methods reduced overlaps of vulnerability areas between sub-models (Yang et al., 2020a).

To further improve the ensembles, mixture-of-experts (MOE) assume that the problem space can be divided into multiple sub-problems through a gate module; the gate module specifies each sub-model on a specific sub-problem (Jacobs et al., 1991; Ma et al., 2018a).

Nevertheless, to the best of our knowledge, MOE-based methods have been not applied to help adversarial robustness. Inspired by MOE, we propose the collaboration framework to defend against adversary attacks.

## 3 COLLABORATION TO DEFEND AGAINST ADVERSARIAL ATTACK

### 3.1 SUPERIORITY OF COLLABORATION

This section shows a *collaboration*, in theory, could make better decisions than an *ensemble*.

**Ensemble** Suppose that there are $M$ learned sub-models $\{f_{\theta_1}, f_{\theta_2}, ..., f_{\theta_M}\}$, given an input $x$, $M$ sub-models make predictions $\{f_{\theta_1}(x), f_{\theta_2}(x), ..., f_{\theta_M(x)}\}$. The ensemble outputs a final prediction ensemble$(x, f_{\theta_1}, ..., f_{\theta_M})$ by the voting-based strategy:

$$\text{ensemble}(x, f_{\theta_1}, ..., f_{\theta_M}) = \arg\max_{y \in \{1,...,K\}} \left( \sum_{i=1}^{M} \mathbb{1}_{y=f_{\theta_i}(x)} \right), \quad (1)$$

where $\mathbb{1}$ is the indicator function. Note that the ensemble outputs the predicted label $y$ that agrees with the majority predictions of the sub-models.

**Definition 1** (best-performing sub-model). *Given an input $x$ and its label $y$, the best-performing sub-model achieves the lowest objective loss on the data $(x, y)$ among all $M$ sub-models:*

$$f_{\theta_{\text{best}}}(x) = \min_{f_{\theta_i} \in \{f_{\theta_1}, .., f_{\theta_M}\}} \ell(f_{\theta_i}(x), y). \quad (2)$$

Note that the best-performing sub-model is w.r.t. the input data $(x, y)$, i.e., different input data correspond to different best-performing sub-models.

**Collaboration** Suppose that there are $M$ learned sub-models $\{f_{\theta_1}, f_{\theta_2}, ..., f_{\theta_M}\}$. Given an input $x$, sub-models make predictions $\{f_{\theta_1}(x), f_{\theta_2}(x), ..., f_{\theta_M(x)}\}$. The collaboration tries to output a final prediction collaboration$(x, f_{\theta_1}, ..., f_{\theta_M})$ by the best-performing sub-model:

$$\text{collaboration}(x, f_{\theta_1}, ..., f_{\theta_M}) = f_{\theta_{\text{best}}}(x). \tag{3}$$

**Proposition 1.** *Given $M$ learned sub-models, the predicted accuracy of the collaboration is upper-bounded that of the ensemble, i.e.,*

$$\mathbb{E}_{(x,y)\in D}\left[\mathbb{1}_{\text{collaboration}(x,f_{\theta_1},...,f_{\theta_M})=y}\right] \geq \mathbb{E}_{(x,y)\in D}\left[\mathbb{1}_{\text{ensemble}(x,f_{\theta_1},...,f_{\theta_M})=y}\right]. \tag{4}$$

*Proof.* Given an $(x, y) \in D$, if the ensemble's prediction is correct, at least one sub-model makes correct prediction, i.e., $\mathbb{1}_{f_{\theta_{\text{best}}}(x)=y}$ holds; therefore, the collaboration' prediction is correct. If the collaboration's prediction is correct, there exists a case that the majority of sub-models make consistent but wrong predictions, while a single sub-model's prediction is correct; then, ensemble's prediction is wrong. Therefore, Proposition 1 holds. □

From Proposition 1, a collaboration can theoretically achieve an equal or higher performance than an ensemble. Next, we will introduce a realization of our collaboration framework.

## 3.2 REALIZATION OF COLLABORATION FOR DEFENDING AGAINST ADVERSARIAL ATTACK

This section realizes the framework of Collaboration for Defending against Adversarial Attack (CDA[2]).

**Notation** We firstly introduce the needed notations. Suppose $\mathcal{X}$ and $\mathcal{Y}$ denote input space and output space, where $\mathcal{Y} = \{1, ..., K\}$ for a $K$-class classification problem. There are $N$ samples in the dataset $D = \{(x, y)\}$, where $x \in \mathcal{X}$ and $y \in \mathcal{Y}$. Let $d_{\inf}(x, x') = \|x - x'\|_\infty$ denotes the infinity distance metric, and $\mathcal{B}_\epsilon[x] = \{x' \in \mathcal{X} \mid d_{\inf}(x, x') \leq \epsilon\}$ is the closed ball of of radius $\epsilon > 0$ centered at $x$. To search for adversarial data within norm ball $\mathcal{B}_\epsilon[x]$, Madry et al. (2018) proposed a projected gradient descent (PGD) method that iteratively searches for adversarial data $\tilde{x}$ ($x$ refers to natural data). $f_\theta(x)$ outputs a K-dimensional predicted probability, i.e., $\hat{\boldsymbol{p}}(x) = [\hat{p}_1(x), ..., \hat{p}_k(x)]$.

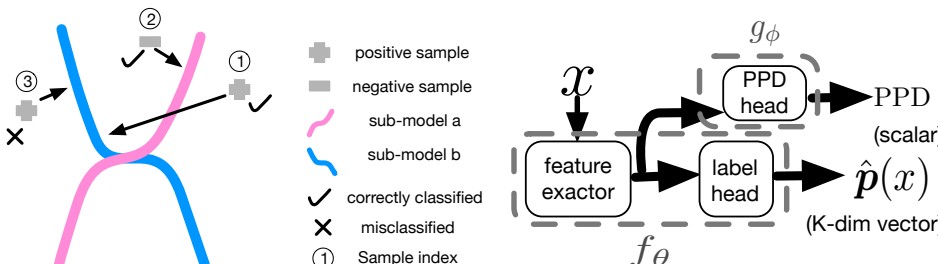

(a) Assign data to their best-performing sub-models          (b) Dual-head structured sub-models

Figure 2: (a) The blue and pink lines denote two sub-models. Each sub-model makes negative predictions $(-)$ on its left and makes positive predictions on its right $(+)$. The given data will be assigned to the sub-model that has the lowest objective loss. The arrows represent the data assignment. (b) Each sub-model has two heads—*label head* that outputs the predicted probability (vector) and *PPD head* that approximates the value of posterior probability density (scalar) of the predicted probability.

**Goal of collaboration** 1) ensure the correct prediction of the best-performing sub-model for a given input, and 2) select the best-performing sub-model among all sub-models to make prediction.

First, intuitively, every sub-model in a collaboration should maximize its expertise to fit its areas and leave the remaining areas fitted by others. As a result, the collaboration can minimize the vulnerability overlaps of all sub-models. Section 4 shows "minimizing the vulnerability overlap of all sub-models" is "minimizing the objective loss of the best-performing sub-models". Therefore, during the training phase, the given data should always be allocated to the sub-model that has the lowest objective loss. In other words, the sub-models always choose the easiest data to learn. In the example of Figure 2(a), $i)$ Data③ is misclassified by both sub-models. The blue sub-model is near Data③ and has the lowest objective loss. We assign the blue sub-model to fit Data③. $ii)$ Data② is correctly classified by the pink model but wrongly classified by the blue model; for ease of effort, we assign the pink model to

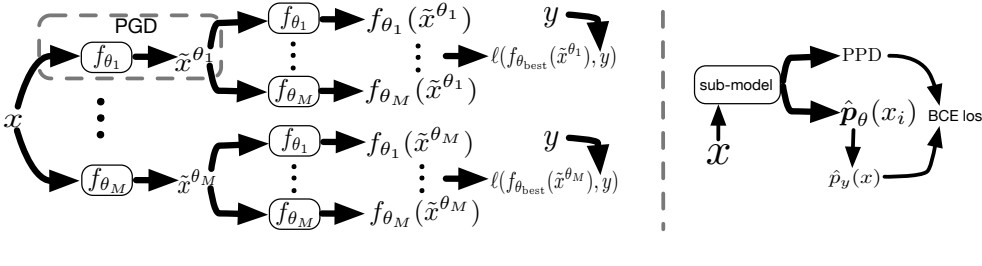

(a) optimizing the module $f_\theta$          (b) optimizing $g_\phi$

Figure 3: Optimization process of the $M$ sub-models in a collaboration

fit Data②, because the collaboration can correctly be classified Data② through selecting the pink model as the representative. $iii$) Data① is correctly classified by both models. The blue model is far from Data① and takes the lowest effort on fitting it; therefore, we assign the blue model to fit Data①.

Second, to select the best-performing sub-model, we construct *dual-head structured sub-models*. As shown in Figure 2(b), our sub-model has dual heads: 1) $f_\theta$ contains the feature extractor module and the label head and predicts the label probability $f_\theta(x) = \hat{\boldsymbol{p}}(x) = [\hat{p}_1(x), ..., \hat{p}_K(x)]$ (a vector); 2) the PPD head $g_\phi$ approximates the *posterior probability density* (PPD) (a scalar) of each prediction $\hat{\boldsymbol{p}}(x)$.

Note that the PPD is the likelihood that a given prediction $\hat{\boldsymbol{p}}(x)$ equals the true label distribution $\boldsymbol{p}(x)$. Thus, the PPD value measures the quality of the sub-model's predictions. In other words, the best-performing sub-models can be decided by the PPD values because the largest PPD value corresponds to the lowest objective loss, and vice versa (see theoretical proof in Proposition 2).

**Collaborate to defend against adversarial attack** To defend against adversarial attack, the collaboration needs to learn from adversarial data. Algorithm 1 along with Figure 3 articulates how to learn such the collaboration, inculding the natural data training and the adversarial data training of the collaboration. Algorithm 2 articulates how to use the collaboration to make predictions. For the natural data training of the collaboration, as shown in Figure 3(a), given natural training data $x$ and its label $y$, we use the PGD method to obtain $M$ adversarial variants $\{\tilde{x}^{\theta_i}\}_{i=1}^M$ of $M$ sub-models as existing baselines do. For each adversarial variant $\tilde{x}^{\theta_i}$, we assign the best-performing sub-model to learn and update its feature extractor and label head. This process corresponds to Lines 2–5 and Lines 9–10 in Algorithm 1. Note that in Line 8, we use a surrogate loss to approximate the sub-model assignment process (reasons see Eq. 6 and 7). As shown in Figure 3(b), given an adversarial variant $\tilde{x}^{\theta_i}$, we propose to use the binary-cross-entropy (BCE) loss between the predicted label probability on the true label (i.e., $\hat{p}_y(\tilde{x}^{\theta_i})$) and the approximated PPD (i.e., $g_\phi(\tilde{x}^{\theta_i})$) to update each sub-model's PPD head. This process corresponds to Lines 6–7 in Algorithm 1.

The natural data training of the collaboration using the most adversarial data of each sub-model may converge without a full exploration of the adversarial samples. For the adversarial data training of the collaboration, we propose to generate the adversarial samples that can worsen the outputs of the collaboration. In particular, for each data sample $x$, we output the prediction $\hat{\boldsymbol{p}}(x)$ whose PPD value $g_\phi(x)$ is the highest. We perturb $x$ to $\tilde{x}$ to worsen the prediction using PGD method. Then we minimize a surrogated loss to fit this adversarial data $\tilde{x}$. This process corresponds to Lines 11-18. In our implementation, the natural data training of the collaboration is not necessary. Because the adversarial data training of the collaboration also captures the most adversarial samples of sub-models and can provide more other adversarial samples for the collaboration.

During the inference phase shown in Algorithm 2, once $M$ sub-models are properly learned, CDA$^2$ chooses a representative sub-model whose PPD value is highest among all sub-models, and then outputs this sub-model's prediction.

### 3.3 ANALYSES OF CDA$^2$

**Optimizing the best-performing sub-models** We firstly show that minimizing the vulnerability overlap of all sub-models is equal to minimizing the objective loss of the best-performing sub-models. For ease of optimization of the best-performing sub-model, we provide a surrogate loss.

---

**Algorithm 1** Collaboration for Defending against Adversarial Attack (CDA$^2$) (training phase)

---

**Input:** sub-models with duel heads $\{f_{\theta_i}\}_{i=1}^M$ and $\{g_{\phi_i}\}_{i=1}^M$, where $f_{\theta_i}$ outputs the label prediction and $g_{\phi_i}$ outputs the approximated PPD, training dataset $D$, hyperparameter $\sigma$;

1:   *the natural data training of the collaboration*
2:   **for** each data $(x, y) \in D$ **do**
3:       **for** each sub-model $f_{\theta_i}, i = 1, 2, ..., M$ **do**
4:          Obtain the adversarial data $\tilde{x}^{\theta_i}$ of the sub-model $f_{\theta_i}$ using PGD method;
5:          **for** each sub-model $f_{\theta_j}, j = 1, 2, ..., M$ **do**
6:             Calculate the approximated PPD, i.e., $g_{\phi_j}(\tilde{x}^{\theta_i})$;
7:             Minimize BCE loss $\ell_\phi = \text{BCE}(g_{\phi_j}(\tilde{x}^{\theta_i}), \hat{p}_y(\tilde{x}^{\theta_i}))$ to update module $g_{\phi_j}$;
8:             Collect sub-model $i$'s cross entropy (CE) loss on data $\tilde{x}^{\theta_i}$: $\ell\left(f_{\theta_j}\left(\tilde{x}^{\theta_i}\right), y\right)$;
9:          Calculate surrogate loss on data $\tilde{x}^{\theta_i}$: $\hat{\ell}_m = -\sigma \ln \sum_{j=1}^M \exp\left(\frac{-\ell\left(f_{\theta_j}(\tilde{x}^{\theta_i}), y\right)}{\sigma}\right)$;
10:         Update $\{f_{\theta_i}\}_{i=1}^M$ by minimizing $\hat{\ell}_m$; // choose the best-performing sub-model to fit $\tilde{x}^{\theta_i}$
11: *the adversarial data training of the collaboration*
12: **for** each data $(x, y) \in D$ **do**
13:       **for** each sub-model $f_{\theta_i}, i = 1, 2, ..., M$ **do**
14:          Calculate the approximated PPD, i.e., $g_{\phi_i}(x)$;
15:          Calculate the prediction i.e., $f_{\theta_i}(x)$;
16:       Output the prediction $\hat{\boldsymbol{p}}'(x)$ with the highest PPD value;
17:       Obtain $\tilde{x}$ by perturbing $x$ to worsen the prediction; // generate the adversarial samples of the collaboration
18:       Minimize BCE loss $\ell_\phi(\tilde{x})$ to update the module $g_\phi$ of all sub-models;
19:       Update $\{f_{\theta_i}\}_{i=1}^M$ to fit $\tilde{x}$ by minimizing the surrogate loss $\hat{\ell}_m(\tilde{x})$;
20: **return** the learned sub-models $\{f_{\theta_i}\}_{i=1}^M$ with $\{g_{\phi_i}\}_{i=1}^M$.

---

The vulnerability overlap of all sub-models refers to the set of adversarial data $(\tilde{x}, y)$ that are misclassified by all sub-models, i.e., all sub-models' objective loss is higher than a certain degree $\delta$:

$$\min_{\theta_i \in \{\theta_1, ..., \theta_M\}} \ell(f_{\theta_i}(\tilde{x}, y)) > \delta, \quad \text{where } \tilde{x}_i \in \tilde{D}, \tag{5}$$

where $\tilde{D}$ denotes the vulnerability overlap of all sub-models.

To reduce the vulnerability overlap of all sub-models, we only need to reduce objective loss of a single model, which is equal to minimizing the loss of the best-performing sub-model, i.e.,

$$\min_{\{\theta_1, \theta_2, ..., \theta_M\}} \mathbb{E}_{(x,y) \in D} \left( \mathbb{E}_{\theta_i \in \{\theta_1, \theta_2, ..., \theta_M\}} \min_{\theta_j \in \{\theta_1, \theta_2, ..., \theta_M\}} \ell\left(f_{\theta_j}(\tilde{x}^{\theta_i}), y\right) \right), \tag{6}$$

where $\tilde{x}^{\theta_i}$ is the adversarial data generated by the sub-model $f_{\theta_i}$.

While directly performing the outer minimization in Eq.(6) may cause a trivial solution (e.g., there is only one optimized sub-model), for ease of the optimization of Eq.(6) (Corresponding to Lines 8–9 in Algorithm 1), we provide a surrogate objective as follows.

$$\min_{\{\theta_1, \theta_2, ..., \theta_M\}} \mathbb{E}_{(x,y) \in D} \left( \mathbb{E}_{\theta_i \in \{\theta_1, \theta_2, ..., \theta_M\}} \hat{\ell}_m(\tilde{x}^{\theta_i}, y) \right), \tag{7}$$

where $\hat{\ell}_m(\tilde{x}^{\theta_i}, y) = -\sigma \ln \sum_{j=1}^M \exp\left(\frac{-\ell\left(f_{\theta_j}(\tilde{x}^{\theta_i}), y\right)}{\sigma}\right)$ and $\sigma > 0$ is a pre-defined hyper-parameter.

In Eq.(7), we approximate the objective $\min_{\theta_j \in \{\theta_1, \theta_2, ..., \theta_M\}} \ell\left(f_{\theta_j}(\tilde{x}^{\theta_i}), y\right)$ using a smooth surrogated maximum function due to

$$\min_{\theta_j \in \{\theta_1, \theta_2, ..., \theta_M\}} \ell\left(f_{\theta_j}(\tilde{x}^{\theta_i}), y\right) - \delta \cdot \ln(M) \leq \hat{\ell}_m(\tilde{x}^{\theta_i}), y) \leq \min_{\theta_j \in \{\theta_1, \theta_2, ..., \theta_M\}} \ell\left(f_{\theta_j}(\tilde{x}^{\theta_i}), y\right). \tag{8}$$

The proof of Eq.(8) is in Appendix.

---

**Algorithm 2** Collaboration for Defending against Adversarial Attack (CDA$^2$) (inference phase)

---

**Input:** the learned sub-models $\{f_{\theta_i}\}_{i=1}^M$ with $\{g_{\phi_i}\}_{i=1}^M$, test input $x$.
1: **for** all sub-models $f_{\theta_i}, i = 1, ..., M$ (in parallel) **do**
2:     make label predictions $\hat{\boldsymbol{p}}(x) = f_{\theta_i}(x)$ and output approximated PPD value $g_{\phi_i}(x)$;
3: **return** prediction $\hat{\boldsymbol{p}}(x)$ whose PPD value $g_\phi(x)$ is the highest among M sub-models.

---

**The best-performing sub-model has the highest PPD.**     We show that a sub-model with the highest PPD achieves the minimum of the objective loss among all sub-models, i.e., the best-performing sub-model.

According to Bayesian theory, for a given input $x$ in a $K$-class classification problem, the true probability of the corresponding label $\boldsymbol{p}(x) = [p_1(x), ..., p_K(x)]$ is agnostic. Usually we assume $\boldsymbol{p}(x)$ comes from a prior probability distribution, e.g., $\boldsymbol{p}(x) \sim \text{Dir}(\boldsymbol{\alpha})$, where $\text{Dir}(\boldsymbol{\alpha})$ is the Dirichlet distribution with the pre-defined parameter vector $\boldsymbol{\alpha} = [\alpha_1, \alpha_2, ..., \alpha_K]$. The prior probability density function of $\text{Dir}(\boldsymbol{\alpha})$ is

$$f(\boldsymbol{p}(x), \boldsymbol{\alpha}) = \frac{1}{\text{B}(\boldsymbol{\alpha})} \prod_{i=1}^K p_i(x)^{\alpha_i - 1} \quad \text{where} \quad \text{B}(\boldsymbol{\alpha}) = \frac{\prod_{i=1}^K \Gamma(\alpha_i)}{\Gamma(\alpha_0)}, \quad \alpha_0 = \sum_{i=1}^K \alpha_i, \quad (9)$$

where $\Gamma(\cdot)$ denotes the Gamma function. According to the Bayesian views, when we learn the model in a supervised manner, the label $y$ happens $n$ times after $n$ observations given the data $(x, y) \in D$. Therefore, the posterior probability distribution is $\text{Dir}(\boldsymbol{\alpha'})$ where $\boldsymbol{\alpha'} = [\alpha_1, ..., \alpha_y + n, ..., \alpha_K]$, and the posterior probability density (PPD) function is

$$\text{PPD}(\boldsymbol{p}(x), \boldsymbol{\alpha'}) = \frac{1}{\text{B}(\boldsymbol{\alpha'})} \prod_{i=1, i \neq y}^K p(x)^{\alpha_i - 1} \cdot p_y(x)^{\alpha_y + n - 1}, \quad (10)$$

where PPD corresponds to the likelihood of a given $\hat{\boldsymbol{p}}(x)$ being the probability of the true label $\boldsymbol{p}(x)$. In other words, the predictions $\hat{\boldsymbol{p}}(x)$ from the sub-model $f_{\theta_i}$ with a higher PPD is more likely to be the true label probability. Therefore, we propose to use PPD to measure the quality of predictions. In Eq.(10), when $n$ (i.e., the number of posterior observations) is sufficiently large, $p_y(x)^{\alpha_y + n - 1}$ becomes a dominant part; therefore, *the value of PPD monotonically increases w.r.t. the value of* $p_y(x)$. Consequently, we have the following proposition.

**Proposition 2.** *Given an input $x$, the sub-model that has the highest PPD corresponds to the best-performing sub-model, i.e.,*

$$\underset{j \in \{1,2,...,M\}}{\arg\max} \text{PPD}(f_{\theta_j}(x), \boldsymbol{\alpha'}) = \underset{j \in \{1,2,...M\}}{\arg\min} \ell(f_{\theta_j}(x), y). \quad (11)$$

From Proposition 2, the PPD value can decide the best-performing sub-model. Given an input to the collaboration, our dual-head structured sub-models can collaboratively decide the best-performing sub-model by just comparing the values of the PPD head (corresponds to Algorithm 2). To learn the PPD head, we could simply compare its output with the predicted probability on the true label (i.e.,$\hat{p}_y(x)$), then update the PPD head by gradient descent (corresponds to Lines 6–7 in Algorithm 1).

Note that the PPD head may be susceptible to adversarial attacks in the white-box setting. In our implementation, we use a simple linear structure to regress the PPD value.

## 4 EXPERIMENTS

In this section, we provide a synthetic experiment to illustrate the behavior of our method compared with prior ensemble methods. Then, we provide a series of experiments on a benchmark dataset to verify the effectiveness of our method in defending against adversarial attacks.

### 4.1 EXPERIMENTAL SETUP

Following the work in (Yang et al., 2020a), we compare our method with various related methods, including ADP (Pang et al., 2019), GAL (Kariyappa & Qureshi, 2019), DVERGE (Yang et al., 2020a). We use ResNet-20 (He et al., 2016) as sub-models in all methods for fair comparisons, and we use CIFAR10 as the data set, a classical image dataset (Krizhevsky et al., 2009) that has 50,000 training images and 10,000 test images.

### 4.2 SYNTHETIC DATA EXPERIMENTS

To demonstrate the behavior of collaboration, we apply our collaboration framework on the well-known **XOR** problem as an example.

In **XOR** problem, there is a binary training set $D = \{x_i, y_i\}_{i=1}^{n}$, where $y_i \in \{\pm 1\}$. For the samples with the label $y_i = 1$, the input feature $x$ is independently sampled from the two Gaussian distributions $x \in \mathcal{N}(\boldsymbol{\mu} = [1,1], \boldsymbol{\sigma} = 0.1 \cdot \boldsymbol{I}_2)$ and $x \in \mathcal{N}(\boldsymbol{\mu} = [-1,-1], \boldsymbol{\sigma} = 0.1 \cdot \boldsymbol{I}_2)$, where $\mathcal{N}(\boldsymbol{\mu} = [1,1], \boldsymbol{\sigma} = 0.1 \cdot \boldsymbol{I}_2)$ denotes 2-d Gaussian distribution with the mean vector $\boldsymbol{\mu} = [1,1]$ and covariance matrix $\boldsymbol{\sigma} = 0.1 \cdot \boldsymbol{I}_2$. For the samples with the label $y_i = -1$, we sample the input feature independently $x_i \in \mathcal{N}(\boldsymbol{\mu} = [-1,1], \boldsymbol{\sigma} = 0.1 \cdot \boldsymbol{I}_2)$ and $x_i \in \mathcal{N}(\boldsymbol{\mu} = [1,-1], \boldsymbol{\sigma} = 0.1 \cdot \boldsymbol{I}_2)$.

Suppose there are two linear sub-models $f_1(x) = \boldsymbol{a}_1 \cdot x + b_1$ and $f_2(x) = \boldsymbol{a}_2 \cdot x + b_2$. Ensemble methods output a prediction $\hat{\boldsymbol{p}}(x)$ by a voting-based strategy e.g., averaging the predictions as the output, i.e., $\hat{\boldsymbol{p}}(x) = \frac{1}{M} \cdot \sum_{i=1}^{M} f_{\theta_i}(x)$.

Collaboration aims to 1). minimize the mean square error (MSE) of the best-performing sub-models; 2). optimize the module $g_1$ and $g_2$ to measure the quality of the multiple predictions during inference. From Figure 4(a), learning multiple linear sub-models and averaging the predictions (ensemble)

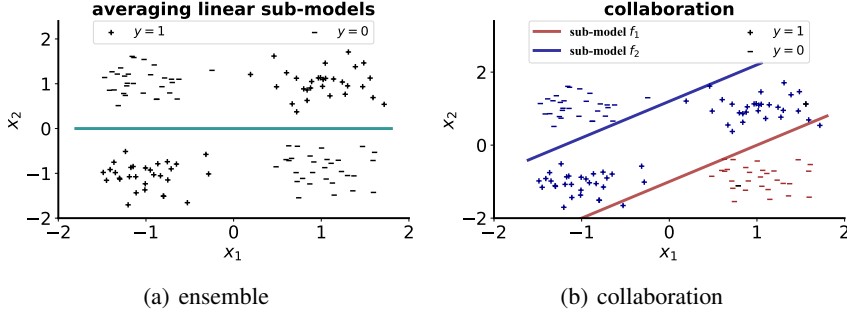

(a) ensemble        (b) collaboration

Figure 4: Illustration of the behavior of the ensemble and the collaboration.

is still a linear model, so it cannot tackle **XOR** problem. Collaboration can address **XOR** which classifies each sample by learning to specify the sub-tasks to different sub-models. From Figure 4(b), the samples are assigned to two sub-models, in which the blue samples are assigned to $f_1$ and the red samples are assigned to $f_2$. Finally, it output the prediction by identifying the best-performing sub-models.

### 4.3 PERFORMANCE ON WHITE-BOX ATTACK

As there are mainly two threat modes in the adversarial attack setting: white-box attack and black-box attack. White-box attack refers to that attackers know all the information about the models, including training data, model architectures, and parameters, while black-box attackers have no access to the information about the model's structures and parameters and rely on surrogate models to generate transferable adversarial examples.

We compare our method with baselines (GAL (Kariyappa & Qureshi, 2019), DVERGE (Yang et al., 2020a) and ADP (Pang et al., 2019)) on defending white-box attacks using a subset of **CIFAR10**. Following in the setting in Yang et al. (2020a), we use 50-step PGD with five random starts and the step size of $\epsilon/5$ to attack all methods. In particular, we randomly select 1000 samples under different $\epsilon$. For the PGD attack, we select the cross-entropy loss to update the perturbations to search for adversarial samples. In addition to the accuracy under different $\epsilon$, we also report the performance of all methods on clean data with the adversarial training in which $\epsilon = 0.03$.

Table 1: Robustness results (%) under white-box attack.

| methods \ $\epsilon$ | clean | 0.01 | 0.02 | 0.03 | 0.04 | 0.05 | 0.06 | 0.07 |
|---|---|---|---|---|---|---|---|---|
| GAL | 81.2 | 49.5 | 31.4 | 25.4 | 22.7 | 18.4 | 13.4 | 9.0 |
| DVERGE | 79.7 | 67.3 | 52.3 | 41.1 | 29.9 | 22.5 | 14.2 | 10.0 |
| ADP | 85.4 | 67.7 | 52.9 | 40.8 | 30.8 | 25.8 | 23.4 | 20.3 |
| $CDA^2$ | 80.2 | **72.0** | **57.5** | **47.8** | **38.7** | **30.4** | **24.3** | **24.0** |

From Table 1, $CDA^2$ achieves a better robustness performance under white-box attack. The results verify that collaboration significantly improves the utilization of the limited model capacity. Therefore, $CDA^2$ can fit more adversarial data and has a relatively smaller vulnerable area.

## 4.4 PERFORMANCE ON BLACK-BOX ATTACK

Due to the transferability of adversarial examples, transfer adversaries can craft adversarial examples based on surrogate models and perform an attack on the target model. In our experiments, we follow the transfer attack setting in (Yang et al., 2020a) and select 1000 test samples randomly. We select 1000 test samples randomly and use hold-out baseline ensembles with three ResNet-20 sub-models as the surrogate models to generate adversarial samples. In particular, we use three attack methodologies: PGD with momentum (Dong et al., 2018), SGM (Wu et al., 2020a) which adds weight to the gradient through the skip connections of the model, and M-FGSM (Xie et al., 2019) which randomly augments the input images in each step. For each sample, three adversarial variants are using the three attack methods. Only when the model can classify all kinds of adversarial variants can the model successfully defend against adversarial attacks. We show the results of all methods in Table 2. In our experiments, GAL is hard to optimize in adversarial training. From Table 2,

Table 2: Robustness results (%) under transfer attack.

| methods \ $\epsilon$ | 0.01 | 0.02 | 0.03 | 0.04 | 0.05 | 0.06 | 0.07 |
|---|---|---|---|---|---|---|---|
| GAL | 65.1 | 49.9 | 49.7 | 47.3 | 53.4 | 51.1 | 42.2 |
| ADP | **85.6** | 83.0 | 79.3 | **79.0** | 69.6 | 60.4 | 57.4 |
| DVERGE | 83.4 | 80.1 | 77.3 | 72.4 | 71.9 | 68.8 | 66.2 |
| $CDA^2$ | 85.4 | **83.4** | 79.3 | 77.0 | **74.2** | **72.3** | **70.2** |

we show the transfer attack robustness of ensemble methods across a wide range of attack radius $0.01 \le \epsilon \le 0.07$. When $0.01 \le \epsilon \le 0.04$, $CDA^2$ achieves a comparable performance compared with the SOTA method. With the increase of $\epsilon$, the volume of $\epsilon$-ball increases exponentially, the performances of all methods get worse significantly because of insufficient model capacity. Since $CDA^2$ addresses more adversarial data using a collaboration mechanism, it achieves a relatively better robustness performance as $0.05 \le \epsilon \le 0.07$.

In addition to transfer attack, query attack can also craft adversarial samples based on the predicted scores of the model. To evaluate the robustness of of $CDA^2$ under query attack, we use Square Attack method (Andriushchenko et al., 2020) to attack all method. Square Attack selects localized square-shaped updates at random positions in each step (Andriushchenko et al., 2020). In particular, we set $\epsilon = 0.03$ and learn the models for each method with adversarial training. Then we evaluate the robustness of each method under Square Attack with 5000 iterations and the results are presented in Table 3. From Table 3, $CDA^2$ outperforms the baselines under query attack. $CDA^2$ optimizes the

Table 3: Robustness results (%) under Square Attack.

| method | GAL | ADP | DVERGE | $CDA^2$ |
|---|---|---|---|---|
| Acc(%) | 25.0 | 47.2 | 48.5 | **53.0** |

utilization of the model capacity by specifying each sub-models to handle the "specific" adversarial attacks, which defends against query attacks more efficiently.

## 5 CONCLUSION

In this paper, we study an essential question in the field of adversarial attacks that when we should collaborate. ($i$) If a single model can handle everything, there is no need for multiple models. ($ii$) If a single model can only handle a part of the whole, collaboration among multiple models makes sense. Adversarial defense is a typical task that falls into the circumstance ($ii$) because a single model hardly fits adversarial data. We provided a collaboration framework—$CDA^2$—as the defense strategy over ensemble methods, and empirical experiments indeed verified the efficacy of $CDA^2$. Future work includes applying our collaboration framework to other areas such as kernel methods, fairness, and global federated model, etc.

ETHICS STATEMENT

In this work, we investigate the robustness issue in deep neural networks (DNNs). We propose a collaboration mechanism for reducing the vulnerability of existing DNNs to enhance their trustworthiness. Having reading the codes of ethics, we make sure that our work conforms to them. In our experiments, we use a public data set which complies with requirements.

REPRODUCIBILITY STATEMENT

Our instructions and experimental settings are illustrated in the maintext. For the theoretical results, we provide rigorous analysis and detailed proofs in maintext and the appendix. For more experimental results and the discussions about CDA[2] , please refer to the Appendix. The source code of our method is attached in Appendix.

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

# A  THEORETICAL PROOF

The proof of Eq.(8) is as follows.

*Proof.* Since we have

$$\max_{j} \exp\left(\frac{-\ell\left(f_{\theta_j}(\tilde{x}^{\theta_i}), y\right)}{\sigma}\right) \leq \sum_{j=1}^{M} \exp\left(\frac{-\ell\left(f_{\theta_j}(\tilde{x}^{\theta_i}), y\right)}{\sigma}\right) \leq M \cdot \max_{j} \exp\left(\frac{-\ell\left(f_{\theta_j}(\tilde{x}^{\theta_i}), y\right)}{\sigma}\right),$$
(12)

considering that $-\delta \ln(x)$ monotonically decreases w.r.t $x$, we have

$$-\delta \ln\left(M \cdot \max_{j} \exp\left(\frac{-\ell\left(f_{\theta_j}(\tilde{x}^{\theta_i}), y\right)}{\sigma}\right)\right) \leq \hat{\ell}_m(f_{\theta_j}(\tilde{x}^{\theta_i}, y)) \leq -\delta \ln \max_{j} \exp\left(\frac{-\ell\left(f_{\theta_j}(\tilde{x}^{\theta_i}), y\right)}{\sigma}\right),$$
(13)

where $-\delta \ln\left(M \cdot \max_j \exp\left(\frac{-\ell\left(f_{\theta_j}(\tilde{x}^{\theta_i}), y\right)}{\sigma}\right)\right) = \min_{\theta_j \in \{\theta_1, \theta_2, ..., \theta_M\}} \ell\left(f_{\theta_j}(\tilde{x}^{\theta_i}), y\right) - \delta \cdot \ln(M)$

and $-\delta \ln \exp\left(\max_j \frac{-\ell\left(f_{\theta_j}(\tilde{x}^{\theta_i}), y\right)}{\sigma}\right) = \min_{\theta_j \in \{\theta_1, \theta_2, ..., \theta_M\}} \ell\left(f_{\theta_j}(\tilde{x}^{\theta_i}), y\right)$, so Eq.(8) holds.   □

The proof of Proposition 2 is as follows.

*Proof.* The objective loss depends on the label probability (e.g., cross-entropy loss $\ell = -\ln p_y(x)$), and the loss monotonically decreases with respect to $p_y(x)$, so the best-performing sub-model has the highest label probability. Combined the fact that $\text{PPD}(\boldsymbol{p}(x))$ monotonically increases w.r.t. $p_y(x)$, the best-performing sub-model corresponds to the highest PPD; therefore, Eq.(11) holds.   □

# B  IMPLEMENTATION DETAILS

**the design of the PPD head**    In our experiments, we use ResNet20 as our backbone model. For the additional PPD head, we propose to use a one-layer MLP to model the relationship between the prediction and its confidence. Our design of the PPD head is based on the following aspects: 1). convenient optimization; a simple model structure can be optimized easily and will not bring a significant computation cost ; 2). robustness; a complex PPD head may be vulnerable to the adversarial attack. Therefore, in our implementation, we choose to learn a simple PPD head for each sub-model to improve its robustness.

**Training devices**    We conduct all experiments on the device GeForce RTX 2080Ti.

# C  TRAINING COST ANALYSIS

Training cost is a notable issue. We compare the training cost of all methods from the two aspects; 1). parameters and GFLOPs: all methods have the same model architecture (ResNet20), so all methods have a similar number of parameters and GFLOps. Compared with baselines, our method has an additional head (PPD head), which is a one-layer MLP with 128 parameters and has a negligible computation cost; 2). training manner; all methods except DVERGE achieve adversarial training by generating adversarial samples using PGD attack. The time consumption of all methods using the device Geforce 2080Ti (100 epochs) is in Table 4.

Table 4: time consumption of all methods (100 epoches).

| methods | GAL | DEVRGE | ADP | CDA$^2$ |
|---------|-----|--------|-----|---------|
| time | 6 h 36min | 11 h 40 min | 6 h 35 min | 7 h 23 min |

DVERGE distills non-robust features by computing transferable adversarial samples, which have a $O(N^2)$ time complexity in which N is the number of sub-models, so it has a relatively large time consumption. Our method outperforms baselines by training an additional PPD head and it could cause an additional small time consumption as shown in the above table.

## D   MORE EXPERIMENTAL RESULTS

For the transfer attack, we also use a more challenging setting following the work in (Yang et al., 2020a). For the attack methods, we use PGD with momentum (Dong et al., 2018) with three random starts, M-FGSM (Xie et al., 2019) and SGM (Wu et al., 2020a). We use hold-out baseline models with 3, 5, and 8 ResNet-20 sub-models as the surrogate models. Meanwhile, we generate adversarial samples with cross-entropy loss and CW loss (Carlini & Wagner, 2017b). For each sample, we generate 30 adversarial variants, and only if the model classifies all the 30 variants can the model defend the transfer attack successfully. The results are shown in the following Table. From Table 5, $CDA^2$ outperforms baselines as $0.01 \leq \epsilon$.

Table 5: Robustness results (%) under transfer attack with 30 adversarial variants.

| methods \ $\epsilon$ | 0.01 | 0.02 | 0.03 | 0.04 | 0.05 | 0.06 | 0.07 |
|---|---|---|---|---|---|---|---|
| GAL | 57.8 | 64.1 | 46.3 | 56.0 | 43.9 | 44.5 | 41.4 |
| ADP | **84.2** | 80.1 | 73.9 | 69.6 | 65.3 | 56.0 | 60.7 |
| DVERGE | 81.5 | 78.1 | 73.5 | 68.4 | 67.2 | **63.8** | 57.1 |
| $CDA^2$ | 83.2 | **80.4** | **75.0** | **71.0** | **69.1** | 62.8 | **61.4** |

## E   MORE DISCUSSION ABOUT THE COLLABORATION

### E.1   MODEL CAPACITY AND COLLABORATION

A model with sufficient capacity to cover all cases does not need to collaborate with others. To verify this claim, we conduct experiments using the ResNet model with different depths and show the clean accuracy (%) of single/multiple models in the following table.

Table 6: the accuracy using different model structures

| depth | 2 | 8 | 14 | 20 |
|---|---|---|---|---|
| single model | 65.0 | 88.3 | 90.5 | 91.9 |
| collaboration | 67.0 | 89.5 | 91.6 | 92.5 |
| gain | 2.0 | 1.2 | 0.9 | 0.6 |

From the Table 6, with the depth 2, the model has the insufficient model capacity to learn the feature extractor, collaboration can have a relatively large improvement (2.0). As the depth of the model is 20, the model has sufficient model capacity to fit all data samples. Collaboration achieves a slight improvement compared with a single model (0.6).

Compared with standard training, adversarial data are adaptively changed based on the current model to smooth the natural data's local neighborhoods. The volume of these surroundings is exponentially large. The model often encounters insufficient model capacity especially when there is a relatively large $\epsilon$ ball. Therefore, it is urgent to improve the utilization of the capacity for adversarial training.

### E.2   COLLABORATION WITH S SINGLE BIG MODEL

**advantages of the collaboration**   Compared with collaboration, a single big model may be difficult to fit the adversarial data. For a big single model with a deeper structure, it may face gradient vanish or gradient explosion during optimization. Without a well-designed optimization method, it could be more vulnerable to an imperceptible perturbation compared to a simple model. The collaboration alleviates this problem by learning multiple relatively small models.

**disadvantages of the collaboration**   A single big model may be a direct solution for addressing insufficient capacity in adversarial training. Compared with learning a single big model, the collaboration needs to design a complicated mechanism to improve the utilization of the model capacity.

### E.3 COLLABORATION IN ADVERSARIAL TRAINING CAN AVOID A TRIVIAL OPTIMIZATION

As our proposed collaboration mechanism enhances the performance of the best-performing sub-model. One may wonder whether it obtains a trivial case, e.g., only one sub-model is properly trained. In fact, our collaboration will not bring such a trivial case. Fitting the adversarial data consumes a tremendous model capacity so the model usually cannot fit all adversarial samples. For a learned sub-model, there still exist adversarial samples that cannot handle. From our proposed collaboration mechanism, these adversarial samples generated from a learned sub-model are more likely to be assigned to other sub-models which perform better. Therefore, in our collaboration, all sub-models will be trained.

Table 7: the accuracy (%) on the clean data

| $CDA^2$ | sub-model A | sub-model B | sub-model A |
|---------|-------------|-------------|-------------|
| 85.6    | 83.9        | 84.2        | 83.9        |

To experimentally verify this claim, we present the accuracies on clean data of all three sub-models with adversarial training ($\epsilon = 0.02$) in the Table 7. From the Table 7, all sub-models have a similar performance on the clean data.

