# OpenReview forum: "Collaborate to Defend Against Adversarial Attacks"
_ICLR.cc/2022/Conference — ICLR 2022 Submitted_

### Official Review · Reviewer_g3Dc · 2021-10-28

**Correctness:** 1
**Technical Novelty And Significance:** 3
**Empirical Novelty And Significance:** 2
**Recommendation:** 3
**Confidence:** 5

**Main Review:**

The idea of adversarial training on each other's adversarial examples is new. However, it is flawed, at least in its current form.
The issue is that such adversarial training may not provide enough coverage and, after the training converges, there may exist adversarial examples that can attack all sub-models.

Consider a hypothetical situation with two sub-models. Both sub-models classify clean images well. However, sub-model A is easily attacked by adding a faint cross pattern near the upper left corner, and sub-model B is easily attacked by adding a faint cross pattern near the lower right corner.

During the proposed training procedure, we'll only encounter these two types of adversarial examples because they are respectively the best attack with lowest Lp epsilon on the two sub-models. Then the two sub-models learn to solve each other's adversarial examples: sub-model A will become robust against faint cross patterns near the lower right corner and sub-model B will become robust against faint cross patterns near the upper left corner. The training procedure therefore converges quickly.

However, after the adversarial training has converged, a vast space of adversarial examples has not been explored at all. There may exist adversarial examples that, although their epsilon is slightly higher than the faint cross patterns, they can attack both models.
The above discussion can be easily extended to the general case of M sub-models. The point is that the proposed adv training procedure can be self-limiting and may not provide enough coverage.


The experimental results are not sufficient to support the claims.
* Standard accuracies on clean images are not reported in Table 1. They are critical missing information.
* The CIFAR-10 model in (Madry et al. 2018) has robust accuracy of 45.8% against L_inf epsilon of 8/255=0.031 with PGD-20. Table 1 shows that the proposed method has robust accuracy of 44.5% against L_inf epsilon of 0.03 with PGD-10. It's unclear that the proposed method has an advantage.
* This paper seems to confuse black-box attacks and transfer attacks. Tables 2 and 3 are transfer attacks. There are no black-box results in this paper.

The role of the PPD head is not explained well.
According to Figure 3(b) and Algorithm 1 line 6, the PPD head is trained by minimizing binary cross entropy between it and the true-label logit from the normal output.
It seems that because the PPD head is trained to also track wrong predictions of a sub-model, it can serve as a confidence score at inference time.
Equations (11)(12)(13) and the surrounding text do not help and they seem irrelevant to the actual implementation.

The philosophical idea claimed in the conclusion section is not new. What's new here is the adversarial training scheme where sub-models train on each other's adversarial examples.


**Summary Of The Paper:**

This paper proposes an ensemble or mixture-of-experts method to defend against adversarial examples. Though the authors prefer to use the term collaboration method to highlight its difference from vanilla ensemble.
The main idea is that, during adversarial training, the sub-models are trained on each other's adversarial examples.

Specifically:
* for each training image, one adversarial example is generated per sub-model by carrying out an attack on each sub-model.
* each adversarial example is (softly) assigned to the sub-model that has the lowest loss on it as a training image.
* each sub-model has a second output called PPD that quantifies its confidence.
* at inference time, for each input, the sub-model with the highest PPD produces the output.

The rationale is that, because each sub-model only needs to cover part of the adversarial example space, they can do a better job.
Experiments on CIFAR-10 with L_inf attacks are reported.

**Summary Of The Review:**

1) Unfortunately the main idea is flawed and the proposed adversarial training may converge without providing enough coverage.
2) The experimental results do not show advantage over previous non-ensemble method.

---

> ### Author Response · Authors · 2021-11-20
> **Response to Reviewer g3Dc (part 1)**
>
> We would like to thank the reviewer for the valuable feedback and we have updated our paper accordingly. Below are our responses to the comments
>
> * To the comments in **Summary Of The Paper** that **This paper proposes an ensemble or mixture-of-experts method to defend against adversarial examples. Though the authors prefer to use the term collaboration method to highlight its difference from the vanilla ensemble.**
>
>     We would like to thank the reviewer for pointing out this issue, and we are very pleased to discuss the relationship between MOE and our work as follows:
>
>     1).**Different from vanilla mixture-of-experts (MOE)** Prior MOE work assumes that the problem space is divided by nature, and the separability of the problem space is irrelevant to the learning model[1]. However, in adversarial training, the adversarial samples depend on the learned model and the distribution of adversarial samples is non-i.i.d. There is a strong correlation between the problem space (defending against adversarial attacks) and the learned model. Existing MOE methods may not be used for defending against adversarial attacks directly.
>
>     2).**The sub-models defend against specific attacks separately** We would like to agree with the reviewer that the sub-models are trained on other's adversarial examples in our method. More importantly, collaboration realizes to defend against adversarial attacks from one sub-model using other sub-models. Specifically, the multiple PPD heads serve as an **attack detector**. Each adversarial attack is assigned to the best-performing sub-model with the highest confidence. In this way, the sub-models in the collaboration do not need to fit the same sample to defend against an adversarial attack compared with ensemble methods.
>
>
> * to the comment about the hypothetical situation proposed by the reviewer **Consider a hypothetical situation with two sub-models...**,
>
>     1). our confusing writing about Algorithm 1 may lead to the misunderstanding that the adversarial training of collaboration is implemented by attacking each sub-model to obtain adversarial samples. Actually, it emphasizes the assignment of input samples, which misses the adversarial training of the collaboration;
>
>     2). in Algorithm 1 we use the adversarial samples of all sub-models to replace the clean samples for fair comparisons with baselines. Meanwhile, as you mentioned, just using the adversarial samples of each sub-model can cause a vast space of adversarial examples not to be explored. Therefore, in addition to the adversarial samples generated from all sub-models directly, the adversarial training of the collaboration is conducted to explore more other adversarial samples (detailed illustration is updated in Algorithm 1);
>
>     3). **the adversarial training of the collaboration can explore other attacks besides the referred two types of adversarial samples.** In the adversarial training of the collaboration, we perturb a sample $x$ to an adversarial sample $\tilde{x}$ so that the collaboration will output a poor prediction with the highest confidence. Suppose the final prediction of $\tilde{x}$ is from sub-model A in the reviewer's hypothetical situation, $\tilde{x}$ will be neither of these two types adversarial samples: 1).assuming $\tilde{x}$ is the faint cross pattern near the lower-right corner, then CDA$^2$ will assign $\tilde{x}$ to sub-model B rather than A, because sub-model B performs better on $\tilde{x}$; 2). assuming $x'$ is the faint cross pattern near the lower-left corner, sub-model A will output a correct prediction because sub-model A is robust to this attack. Therefore, the adversarial training of the collaboration can explore other attacks besides the most adversarial samples of the two sub-models.
>
>     We would like to thank the reviewer for pointing out this issue and we have updated our submission accordingly.
>
> [1] Robert A Jacobs, Michael I Jordan, Steven J Nowlan, and Geoffrey E Hinton. Adaptive mixtures of local experts. Neural computation 1991.

---

> ### Author Response · Authors · 2021-11-20
> **Response to Reviewer g3Dc (part 2)**
>
> * to the comment about the experiments,
>
>     **Question 1: Standard accuracies on clean images are not reported in Table 1. They are critical missing information.**
>
>     We would like to thank the reviewer for pointing out this mistake and we have updated our paper accordingly. The model is learned using adversarial training with $\epsilon = 8/255$ and the standard accuracies on clean data is as follows:
>
>     | method | GAL | DVERGE | ADP | $\mathrm{CDA}^{2}$ |
>     | ---- | ---- | ---- | ---- | ---- |
>     | **Acc($\\%$)** | 81.2% | 79.7% | 85.4% | 80.2% |
>
>     **Question 2: The CIFAR-10 model in (Madry et al. 2018) has robust accuracy of 45.8% against L_inf epsilon of 8/255=0.031 with PGD-20. Table 1 shows that the proposed method has robust accuracy of 44.5% against L_inf epsilon of 0.03 with PGD-10. It's unclear that the proposed method has an advantage.**
>
>     We would like to acknowledge that our method is evaluated without a careful optimization of the hyper-parameters. Meanwhile, for fair comparisons, we compare our method with existing related baselines (ADP [3], GAL [4], and DVERGE [2]) under the setting that is different from the setting in (Madry et al. 2018). To verify the effectiveness of our work under white-box attack, we compare our method with baselines using PGD-50 attack in the setting in [2]. The results are presented in the following table.
>
>     | $\epsilon$ |  0.01 | 0.02 | 0.03 | 0.04 | 0.05 | 0.06 | 0.07 |
>     | ---- | ----  | ---- | ---- | ---- | ---- | ---- | ---- |
>     | GAL |  49.5 | 31.4 | 25.4 | 22.7 | 18.4 | 13.4 | 9.0 |
>     | DVERGE |  67.3 | 52.3 | 41.1 | 29.9 | 22.5 | 14.2 | 10.0 |
>     | ADP |  67.7 | 52.9 | 40.8 | 30.8 | 25.8 | 23.4 | 20.3 |
>     | $\mathrm{CDA}^{2}$ |  **72.0** | **57.5** | **47.8** | **38.7** | **30.4** | **24.3** | **24.0** |
>
>    From the white-box results above, our method achieves a better robustness performance under white-box attack compared with baselines. The results verify that the collaboration improves the utilization of the limited model capacity. Therefore, CDA$^2$ can fit more adversarial data and has a relatively smaller vulnerable area.
>
>    **Question 3: This paper seems to confuse black-box attacks and transfer attacks. Tables 2 and 3 are transfer attacks. There are no black-box results in this paper.**
>
>     We would like to thank the reviewer for pointing out this issue. There are two types of black-box attacks: transfer attack and query attack. For the transfer attack, we design two settings: (1). generating 3 adversarial variants using PGD with momentom[6], SGM[7], and M-FGSM[8]; (2). generating 30 adversarial variants using the ensemble with the different number of sub-models, various attack methods and different loss functions.
>
>     For each sample, only when the model can classify all kinds of adversarial variants can the model successfully defend against this attack.
>
>     In setting (1), the experimental results are presented in the following table.
>
>     | $\epsilon$ | 0.01 | 0.02 | 0.03 | 0.04 | 0.05 | 0.06 | 0.07 |
>     | ---- | ---- | ---- | ---- | ---- | ---- | ---- | ---- |
>     | GAL | 65.1 | 49.9 | 49.7 | 47.3 | 53.4 | 51.1 | 42.2 |
>     | DVERGE | 83.4 | 80.1 | 77.3 | 72.4 | 71.9 | 68.8 | 66.2 |
>     | ADP | **85.6** | 83.0 | 79.3 | **79.0** | 69.6 | 60.4 | 57.4 |
>     | $\mathrm{CDA}^{2}$ | 85.4 | **83.4** | **79.3** | 77.0 | **74.2** | **72.3** | **70.2** |
>
>     In the setting (2), we show the results in the following table.
>     | $\epsilon$ | 0.01 | 0.02 | 0.03 | 0.04 | 0.05 | 0.06 | 0.07 |
>     | ---- | ---- | ---- | ---- | ---- | ---- | ---- | ---- |
>     | GAL | 57.8 | 64.1 | 46.3 | 56.0 | 43.9 | 44.5 | 41.4 |
>     | DVERGE | 81.5 | 78.1 | 73.5 | 68.4 | 67.2 | **63.8** | 60.7 |
>     | ADP | **84.2** | 80.1 | 73.9 | 69.6 | 65.3 | 56.0 | 51.9 |
>     | $\mathrm{CDA}^{2}$ | 83.2 | **80.4** | **75.0** | **71.0** | **69.1** | 62.8 | **61.4** |
>
>     From the tables above, our method achieves better performance compared with existing methods when there is a relatively large $\epsilon$. As the $\epsilon$ increases, all methods cannot fit enough adversarial samples. The collaboration focuses on minimizing the vulnerability overlap of all sub-models and achieves relatively better performance when $0.01 \leq \epsilon$.

---

> ### Author Response · Authors · 2021-11-20
> **Response to Reviewer g3Dc (part 3)**
>
> * to the comments about experiments,
>
>     Following your suggestion, we also validate the effectiveness of our method in defending against query attack. We train models for each method with adversarial training where $\epsilon = 0.03$. The experimental results of query attack (5000 queries using Square attack [5]) are reported in the following table.
>
>     | method | GAL | DVERGE | ADP | $\mathrm{CDA}^{2}$ |
>     | ---- | ---- | ---- | ---- | ---- |
>     | **Accuracy** |25.0% | 48.5% | 47.2% | **53.0%** |
>
>    Compared with existing ensemble methods, CDA$^{2}$ optimizes the utilization of the model capacity by specifying each sub-models to defend against the "specific" adversarial attacks, which minimizes the vulnerability overlap of all sub-models and improves the robustness of the collaboration when faced with query attacks.
>
> * **The role of the PPD head is not explained well. According to Figure 3(b) and Algorithm 1 line 6, the PPD head is trained by minimizing binary cross-entropy between it and the true-label logit from the normal output.**
>
>     As you understand, the PPD head is trained by minimizing the binary cross-entropy between it and the predicted logit of the label class. It tracks the predictions of the sub-model whether they are correct. Roughly speaking, collaboration detects adversarial attacks before defending against them. Because during training and inference procedure, the multiple ppd heads serve as an **attack detector** and each sample is assigned to the sub-model that performs best to minimize the vulnerability overlap of all sub-models as shown in Figure 1.
>
>     Equations 11-13 explain that the PPD value is related to the likelihood of the corresponding predictions being the true label from a statistical point of view. Meanwhile, the best-performing sub-model we aim to enforce has the highest PPD value as illustrated in Proposition 2.
>
>
> [1] Robert A Jacobs, Michael I Jordan, Steven J Nowlan, and Geoffrey E Hinton. Adaptive mixtures of local experts. Neural computation 1991.
>
> [2] Dverge: Diversifying vulnerabilities for enhanced robust generation of ensembles. In NeurIPS, 2020a.
>
> [3] Improving adversarial robustness via promoting ensemble diversity. In ICLR, 2019.
>
> [4] Improving adversarial robustness of ensembles with diversity training. In Arxiv.
>
> [5] Square Attack: a query-efficient black-box adversarial attack via random search In ECCV, 2020.
>
> [6] Boosting adversarial attacks with momentum. ECCV 2018.
>
> [7] Skip connections matter: On the transferability of adversarial examples generated with resnets. ICLR 2020
>
> [8] Improving transferability of adversarial examples with input diversity. CVPR 2019.

---

> ### Author Response · Authors · 2021-11-24
> **Need further clarification?**
>
> Thanks very much for your constructive comments on our work. We have tried our best to address the concerns. Is there any unclear point so that we should/could further clarify?

---

> ### Comment · Reviewer_g3Dc · 2021-11-25
> **two comments after author response**
>
> First. Comparing the original submission and the new version, there are major changes to the proposed method. The second half of Algorithm 1 is new and an extra paragraph is added to page 5 to explain the new half of Algorithm 1.
>
> The author's response states that "our confusing writing about Algorithm 1 may lead to the misunderstanding that...", which indicates that the new Algorithm 1 is the proposed method all along and the original submission was a misrepresentation.
>
> However, I am unable to find contents in the original submission that hint at the new half of Algorithm 1. To the contrary, there are contents in the original submission that contradict the new half of Algorithm 1. For example, on page 2 of the original submission, there was a statement of "...the M sub-models generate a set of M adversarial variants...", which is deleted in the new version.
>
> I find it difficult to reconcile these observations with the claim that the new Algorithm 1 is the proposed method all along.
>
> Second. In response to AC's question on transfer-attack accuracy, the authors clarified that each column in Table 2 is a different model trained with a different epsilon.
>
> This brings up a question about Table 1. Is each column in Table 1 also  a different model trained with a different epsilon?
> * If the answer is yes. Then Table 1 is still missing critical information. Each different model should have its standard accuracy reported. The "clean" column should be removed and each column should have two numbers.
> * If the answer is no. Then Table 1 and Table 2 are misleading. They have almost identical format yet very different methodology of comparison.

---

> > ### Author Response · Authors · 2021-11-25
> > **Response to the two comments**
> >
> > * to the comment **First,**
> >
> >     firstly, **we have to apologize for our misleading statement "our confusing writing about Algorithm 1 may lead to the misunderstanding that...".**
> >
> >     We would like to acknowledge our mistake that **our original submission missed the information about the adversarial training of the collaboration**
> >
> >     For the baselines, e.g., ADP, the adversarial samples are generated in two ways: (1). generating adversarial samples for each sub-models directly; (2). generating adversarial samples for the ensemble (the ensemble means averaging all sub-models). In our original experiments, we actually followed the baselines and generated adversarial samples in these two ways. However, **in our original submission, we did not realize that without (2) the collaboration may converge without providing enough coverage until the reviewer pointed it out.** Therefore, we just showed the first half of our method in the original Algorithm 1. Now, we have corrected our writings, the current Algorithm is correct.
> >
> >
> >     We would like to appreciate the reviewer pointing out the mistaken writing. Thanks to the policy of ICLR, we have the opportunity to revise our submission accordingly.
> >
> >
> > * to the comment **Second**, the answer is **yes**. Each column in Table 1 also uses a different model trained with a different $\epsilon$.
> >
> >     We thought that you mean we missed the clean accuracy with $\epsilon = 8/255 \approx 0.031$ and we highlighted it in **Question 1**.
> >
> >     Following your suggestion, we remove the "clean" accuracy in Table 1. In the following table, we report the robust/clean accuracy for each model.
> >
> >  | $\epsilon$ (robust/clean) | 0.01| 0.02 | 0.03 | 0.04 | 0.05 | 0.06 | 0.07|
> >  | ----  | ---- | ---- | ---- | ---- | ---- | ---- | ---- |
> >  | GAL | 49.5/87.8 | 31.4/85.4 | 25.4/81.2 | 22.7/78.7 | 18.4/77.3 | 13.4/76.2 | 9.0/**76.0** |
> >  | DVERGE | 67.3/85.4 | 52.3/83.0 | 41.1/79.7 | 29.9/77.6 | 22.5/76.7 | 14.2/75.8 | 10.0/75.3 |
> >  | ADP  | 67.7/**89.0** | 52.9/**86.8** | 40.8/**85.4** | 30.8/**83.3** | 25.8/76.0 | 23.4/66.4 | 20.3/63.0 |
> >  | $\mathrm{CDA}^{2}$ | 72.0/88.8 | 57.5/85.6 | 47.8/80.2 | 38.7/80.0 | 30.4/**79.1** | 24.3/**76.7** | 24.0/74.1 |

---

### Official Review · Reviewer_Koh8 · 2021-11-02

**Correctness:** 3
**Technical Novelty And Significance:** 4
**Empirical Novelty And Significance:** 4
**Recommendation:** 8
**Confidence:** 4

**Details Of Ethics Concerns:**

Nothing.

**Main Review:**

# Strength:

1. This work considers the adversarial defense problem. It focuses on the insufficient model capacity of adversarial training and presents a completely fresh perspective on learning multiple models to improve the robustness. The proposed framework is technically solid. The idea of collaboration to minimize the vulnerable area is innovative compared with baselines. In summary, I assess a high novelty of this work;

2. The proposed framework CDA^{2} makes sense for the goal of minimizing the vulnerable area of the overlap. A dual-head model is used for predicting and providing information about how to assign the adversarial sample. Optimizing the best-performing sub-model is to minimize the vulnerability overlap of all sub-models. The framework has contributed to the research problem and may enlighten other research problems. It could be a good paper for the ICLR community.

3. This work is well-written. Figs. 1 and 2 show the motivation of the proposed method clearly, which helps understand it.

# Weakness:

1. the author states that the insufficient model capacity can hurt its performance in adversarial training. What if the model has sufficient model capacity? Are there other scenarios that have insufficient model capacity? Whether CDA^{2} can be useful for such scenarios if any? Compared to a big single model, can the author discuss more pros and cons of collaboration?

2. To achieve collaboration, the author proposes to assign the samples to the sub-models that perform best. Does it obtain a trivial case? For example, only one sub-model has been trained. Can the author discuss more?

3. For the framework CDA^{2}, ppd head is for evaluating the performance of the other head. Can the author discuss the impact on the quality of this head more?

4. The experiment on the XOR problem is a little bit weak. Can the author provide more experiments to verify the effectiveness of CDA^{2}?

5. For the experimental results on the white-box, the author gives the results in Table 1. The experimental setting is a little vague. I’m wondering about the robustness performance of PGD-50 as in [1];

6. For the black-box experiments, the author uses M-FGSM and PGD to generate transferable adversarial samples. Can the author provide more experimental results to validate its claims?

Overall, there are still some issues stated above. I will increase my score if they are well addressed.

[1]. Dverge: Diversifying vulnerabilities for enhanced robust generation of ensembles. In NeurIPS, 2020a.

========

# Post rebuttal responses:

Thanks for the authors' responses. They address my concerns on the concept of collaboration, and the newly added experiments are convincing. Especially, I like the idea of collaboration in adversarial training, which is a new paradigm to defend against adversarial attacks.

Besides robustness, the new paradigm may be helpful to other domains.

After reading the review from other reviewers and the corresponding responses, I vote for acceptance and increase my scores further.

**Summary Of The Paper:**

This paper firstly analyzes prior adversarial defense methods using ensemble strategy and claims that this method could cause a waste of model capacity. To improve the utilization of (multiple ) model capacity, the author proposes an interesting collaboration strategy $CDA^2$ to defend against adversarial attacks. Specifically, the author develop a dual-head model structure: one is for making a prediction and the other is for predicting the posterior probability of the input. During training, each model can address the adversarial attacks of other sub-models so that it improves the robustness of the collaboration. The experimental results partially verify the superiority of the proposed methods.

**Summary Of The Review:**

A novel and interesting collaboration method for advancing the robustness of multiple sub-models.

---

> ### Author Response · Authors · 2021-11-20
> **Response to Reviewer Koh8 (part 1)**
>
> We would like to thank the reviewer for recognizing the contribution of our work and the insightful feedback. We have revised our submission accordingly and below are our responses to the comments.
>
> * to the first comment in **Weakness**,
>
>     **Question 1 : What if the model has sufficient model capacity?**
>
>     **Answer:** We would like to explain that a model with sufficient capacity to cover all cases does not need to collaborate with others. To verify this claim, we conduct experiments using the ResNet model with different depths and show the clean accuracy ($\%$) of single/multiple models in the following table.
>
>     | Depth | 2 | 8 | 14 | 20 |
>     | ---- | ---- | ---- | ---- | ---- |
>     | single model | 65.0 | 88.3 | 90.5 | 91.9 |
>     | Collaboration | 67.0 | 89.5 | 91.6 | 92.5 |
>     | Gains | 2.0 | 1.2 | 0.9 | 0.6 |
>
>     From the table above, with the depth 2, the model has the insufficient model capacity to learn the feature extractor, collaboration can have a relatively large improvement (2.0). As the depth of the model is 20, the model has sufficient model capacity to fit all data samples. Collaboration achieves a slight improvement compared with a single model (0.6).
>
>     Compared with standard training, adversarial data are adaptively changed based on the current model to smooth the natural data’s local neighborhoods. The volume of these surroundings is exponentially large. The model often encounters insufficient model capacity especially when there is a relatively large $\epsilon$ ball. Therefore, it is urgent to improve the utilization of the capacity for adversarial training.
>
>     **Question 2: Are there other scenarios that have insufficient model capacity? Whether $CDA^{2}$ can be useful for such scenarios if any?**
>
>     **Answer:**  we are very pleased to share our thoughts on this open question. There may be other scenarios in addition to the robustness issue. Take the fairness issue for an example, prior works on fairness reveal that there may exist a fairness-accuracy trade-off [1]. Therefore, under the fairness constraint, the model may not achieve a better trade-off due to insufficient capacity.
>
>     Collaboration may shed light on this problem. To learn a fair representation, it may be helpful to assign each sample to the sub-model that achieves a better fairness-utility trade-off on it. Collaboration may have a better performance Because each sub-model is specified to deal with the samples it is expert in. More detailed designs for this problem need further exploration and we will continue to study this problem.
>
>     **Question 3: Compared to a big single model, can the author discuss more pros and cons of collaboration?**
>
>     **Answer:** For this question, we would like to share our views as follows.
>
>     1). **pros :** a single big model may be difficult to fit the adversarial data. For a big single model with a deeper structure, it may face gradient vanish or gradient explosion during optimization. Without a well-designed optimization method, it could be more vulnerable to an imperceptible perturbation compared to a simple model. Compared with optimizing a single big model, the collaboration alleviates this problem by learning multiple relatively small models.
>
>     2). **cons :** A single big model may be a direct solution for addressing insufficient capacity in adversarial training. Compared with learning a single big model, the collaboration needs to design a complicated mechanism to improve the utilization of the model capacity.
>
>
> * **To achieve collaboration, the author proposes to assign the samples to the sub-models that perform best. Does it obtain a trivial case? For example, only one sub-model has been trained. Can the author discuss more?**
>
>     The collaboration will not obtain a trivial case because of the adversarial samples. Fitting the adversarial data consumes a tremendous model capacity, so the model usually cannot fit all adversarial samples. For a learned sub-model, there still exist adversarial samples it cannot fit. From our proposed collaboration mechanism, these adversarial samples generated from a learned sub-model are more likely to be assigned to other sub-models which perform better. Therefore, in our collaboration, all sub-models will be trained.
>
>     | Collaboration | sub-model A | sub-model B | sub-model C |
>     | --- | --- | --- | --- |
>     | 85.6 | 83.9 | 84.2 | 83.9 |
>
>     To experimentally verify this claim, we present the accuracies on clean data of all three sub-models with adversarial training ($\epsilon = 0.02$) in the above table. From the table, all sub-models have a similar performance on the clean data.
>
> [1] Inherent Tradeoffs in Learning Fair Representations. In NeurIPS 2019

---

> ### Author Response · Authors · 2021-11-20
> **Response to Reviewer Koh8 (part 2)**
>
> * **For the framework CDA^{2}, ppd head is for evaluating the performance of the other head. Can the author discuss the impact on the quality of this head more?**
>
>     As shown in Figure 2 (b), our sub-model has two heads: head A is for predicting the label of the input image; head B is for measuring the quality of the prediction from head A. In our implementation, we use a simple neural network to construct the PPD head based on two aspects: 1). convenient optimization; a simple model structure can be optimized easily and will not bring a significant computation cost; 2). robustness; a complex PPD head may be vulnerable to the adversarial attack. Therefore, in our implementation, we choose to learn a simple PPD head for each sub-model to improve its robustness.
>
> * **The experiment on the XOR problem is a little bit weak. Can the author provide more experiments to verify the effectiveness of CDA^{2}?**
>
>     We would like to thank the reviewer for the kind suggestion. As you mentioned, XOR is a simple problem and we use it to show the motivation behind the collaboration. To further verify the effectiveness of our method, we compare our method with baselines under white-box attack, transfer attack and query attack in the following. We also revise our paper accordingly.
>
> * **For the experimental results on the white-box, the author gives the results in Table 1. The experimental setting is a little vague. I’m wondering about the robustness performance of PGD-50 as in [2]**
>
>  We would like to thank the reviewer for the kind suggestions. Following the advice, we conduct the comparisons with baselines under PGD-50 attacks as the setting in [2]. The results are reported in the following table.
>
>  | $\epsilon$ | clean | 0.01 | 0.02 | 0.03 | 0.04 | 0.05 | 0.06 | 0.07 |
>  | ---- | ---- | ---- | ---- | ---- | ---- | ---- | ---- | ---- |
>  | GAL | 81.2 | 49.5 | 31.4 | 25.4 | 22.7 | 18.4 | 13.4 | 9.0 |
>  | DVERGE | 79.7 | 67.3 | 52.3 | 41.1 | 29.9 | 22.5 | 14.2 | 10.0 |
>  | ADP | 85.4 | 67.7 | 52.9 | 40.8 | 30.8 | 25.8 | 23.4 | 20.3 |
>  | $\mathrm{CDA}^{2}$ | 80.2 | **72.0** | **57.5** | **47.8** | **38.7** | **30.4** | **24.3** | **24.0** |
>
>
>  From the white-box results above, our method achieves a better robustness performance under white-box attack compared with baselines. The results verify that collaboration significantly improves the utilization of the limited model capacity. Therefore, CDA$^2$ can fit more adversarial data and has a relatively smaller vulnerable area.
>
> * **For the black-box experiments, the author uses M-FGSM and PGD to generate transferable adversarial samples. Can the author provide more experimental results to validate its claims?**
>
>  We would like to thank the reviewer for the suggestion. To validate the effectiveness of our method when faced with transfer attacks, we design 3 different transfer adversarial variants for each sample using PGD with momentum[3], SGM[4] and M-FGSM[5]. For each sample, only when the model can classify all kinds of adversarial variants can the model successfully defend against adversarial attacks. The results are reported in the following table.
>
>  | $\epsilon$ | 0.01 | 0.02 | 0.03 | 0.04 | 0.05 | 0.06 | 0.07 |
>  | ---- | ---- | ---- | ---- | ---- | ---- | ---- | ---- |
>  | GAL | 65.1 | 49.9 | 49.7 | 47.3 | 53.4 | 51.1 | 42.2 |
>  | DVERGE | 83.4 | 80.1 | 77.3 | 72.4 | 71.9 | 68.8 | 66.2 |
>  | ADP | **85.6** | 83.0 | 79.3 | **79.0** | 69.6 | 60.4 | 57.4 |
>  | $\mathrm{CDA}^{2}$ | 85.4 | **83.4** | **79.3** | 77.0 | **74.2** | **72.3** | **70.2** |
>
> To further verify the effectiveness of our method when faced with transfer attack, we also evaluate our method in a more challenging setting in [2]. We design 30 adversarial variants for each sample. Similarly, only when the model can classify all kinds of adversarial variants can the model successfully defend against adversarial attacks. The results are reported in the following table.
>
>  | $\epsilon$ | 0.01 | 0.02 | 0.03 | 0.04 | 0.05 | 0.06 | 0.07 |
>  | ---- | ---- | ---- | ---- | ---- | ---- | ---- | ---- |
>  | GAL | 57.8 | 64.1 | 46.3 | 56.0 | 43.9 | 44.5 | 41.4 |
>  | DVERGE | 81.5 | 78.1 | 73.5 | 68.4 | 67.2 | **63.8** | 60.7 |
>  | ADP | **84.2** | 80.1 | 73.9 | 69.6 | 65.3 | 56.0 | 51.9 |
>  | $\mathrm{CDA}^{2}$ | 83.2 | **80.4** | **75.0** | **71.0** | **69.1** | 62.8 | **61.4** |
>
>  From the table above, our method achieves a comparable performance compared with existing methods. As the $\epsilon$ increases, all methods cannot fit enough adversarial samples. The collaboration focuses on minimizing the vulnerability overlap of all sub-models and achieves relatively better performance when $0.01 \leq \epsilon$.

---

> ### Author Response · Authors · 2021-11-20
> **Response to Reviewer Koh8 (part 3)**
>
> [1] Inherent Tradeoffs in Learning Fair Representations. In NeurIPS 2019
>
> [2] Dverge: Diversifying vulnerabilities for enhanced robust generation of ensembles. In NeurIPS, 2020a.
>
> [3] Boosting adversarial attacks with momentum. ECCV 2018.
>
> [4] Skip connections matter: On the transferability of adversarial examples generated with resnets. ICLR 2020
>
> [5] Improving transferability of adversarial examples with input diversity. CVPR 2019.

---

### Official Review · Reviewer_qeVf · 2021-11-03

**Correctness:** 4
**Technical Novelty And Significance:** 3
**Empirical Novelty And Significance:** 3
**Recommendation:** 6
**Confidence:** 3

**Main Review:**

Strengths
1. The proposed method is clearly motivated and defined. And it has been demonstrated effective by quantitative experimental results.
2. A comprehensive overview of related works is provided.

Weaknesses
1. Training cost is also a notable issue for adversarial training methods. It would be better if the training cost of the proposed method is compared with that of the previous methods.

**Summary Of The Paper:**

This paper presents a new paradigm for defending against adversarial attacks with multiple sub-models.
Different from ensemble, the proposed collaboration paradigm, a representative sub-model is chosen to make the decision, instead of letting all sub-models vote.
The proposed method has been validated on CIFAR-10 dataset, against both white-box and transferrability-based black-box attacks.

**Summary Of The Review:**

This paper presents a new paradigm involving multiple sub-models which has several advantages compared to ensemble.
It has been validated effective by the quantiative results in terms of robustness, but comparison in other aspects such as training cost could be also helpful.

---

> ### Author Response · Authors · 2021-11-20
> **Response to Reviewer qeVf**
>
> We would like to thank the reviewer for acknowledging the significance of our work and providing the valuable feedback, and we have updated our submission accordingly. Below is our response to the comment.
>
> * **Question: Training cost is also a notable issue for adversarial training methods. It would be better if the training cost of the proposed method is compared with that of the previous methods.**
>
>     **Answer:** we acknowledge that training cost is a notable issue and we would like to thank the reviewer for reminding us of it. Our method will not bring a significant training cost and we compare the training cost of all methods from the two aspects;
>
>     1). **parameters and GFLOPs:** all methods have the same model architecture (ResNet20), so all methods have a similar number of parameters and GFLOps. Compared with baselines, our method has an additional head (PPD head), which is a one-layer MLP with 128 parameters and has a negligible computation cost;
>
>     2). **training manner:** all methods except DVERGE achieve adversarial training by generating adversarial samples using PGD attack.
>
>     | method | GAL | DVERGE | ADP | $\mathrm{CDA}^{2}$ |
>     | ---- | ---- | ---- | ---- | ---- |
>     | **time** | 6 h 36min | 11 h 40 min | 6 h 35 min | 7 h 23 min |
>
>     We report the time consumption of all methods using the same device (100 epochs) in the above table. DVERGE distills non-robust features by computing transferable adversarial samples, which have a O($N^2$) time complexity in which N is the number of sub-models, so it may have a relatively large time consumption. Our method outperforms baselines by training an additional PPD head and it could cause an additional small time consumption as shown in the above table.

---

### Author Response · Authors · 2021-11-20
**Thank the three anonymous reviewers**

We would like to thank the three anonymous reviewers for the very valuable feedback and the kind suggestions. We have uploaded our source code and revised our submission accordingly;

* **for Reviewer qeVf**, we discuss the training cost of all methods in our Appendix;
* **for Reviewer Koh8**, we present our discussions about CDA$^{2}$ in our Appendix. We show more experimental results in Sec 4 (including white-box, transfer attack, and query attack). More experimental results can be found in Appendix;
* **for Reviewer g3Dc**, we clarify the adversarial training of the collaboration for generating more other adversarial samples in Sec 3.2 and we added it in Algorithm 1. To verify the effectiveness of our method experimentally, we present the experimental result on PGD-50 attack in Sec 4.2. We give the transfer attack and query attack experimental results in Sec 4.3. More experimental results are shown in Appendix.

Our responses to the main concerns are given to each reviewer separately. Moreover, we want to know whether there are **unresolved or new concerns** we need to clarify and we are very pleased to discuss them further, if any.

Thank you again!

---

### Decision · Program_Chairs · 2022-01-20

**Decision:**

Reject

**Comment:**

The paper proposes a novel ensemble method, CDA^2,  in which base models collaborate to defend against adversarial attacks. To do so the base models have two heads: the label head for predicting the label and the posterior probability density (PPD) head that is trained by minimizing binary cross entropy between it and the true-label logit given by the label head. During inference the base model with the highest PPD value is chosen to make the prediction. During training base models learn from the adversarial examples produced by other base models.

The evaluation of the manuscript of different reviewers was very diverse, resulting in final scores ranging between 3 and 8 after the discussion period. While the rebuttal clearly addressed the concerns of one reviewer and several additional experimental results were added for different adversarial attacks, it did not fully addressed the concerns of another reviewer, who rated his confidence higher. He was also not convinced by the update in the revised version of the manuscript, in which crucial changes in the pseudocode describing the proposed algorithm were made, which contradicted some statements in the first version. Therefore, the paper can unfortunately not be accepted in its current version. In a future version of the manuscript, the description of the algorithm and of he role of the PPD head should be improved and experiments on another dataset next to CIFAR-10 could be added.